# Prevalence of childhood stunting and determinants in low and lower-middle income African countries: Evidence from standard demographic and health survey

Tadesse Tarik Tamir [1] *, Soliyana Adisu Gezhegn [2], Dejen Tegegne Dagnew [2], Abebe Tilahun Mekonenne [2], Genetu Tadese Aweke [2], Ayenew Molla Lakew [3]

1 Department of Pediatrics and Child Health Nursing, College of Medicine and Health Sciences, University of Gondar, Gondar, Ethiopia, 2 Department of Epidemiology, Institute of Public Health, College of Medicine and Health Sciences, University of Gondar, Gondar, Ethiopia, 3 Department of Epidemiology and Biostatistics, Institute of Public Health, College of Medicine and Health Sciences, University of Gondar, Gondar, Ethiopia

* tadestar140@gmail.com

## Abstract

**Data Availability Statement:** The data used for analysis of the current study are available publicly online at the MEASURE DHS program [https://

## Introduction

Undernutrition poses a significant global public health challenge, adversely affecting childhood cognitive and physical development while increasing the risk of disease and mortality. Stunting, characterized by impaired growth and development in children due to insufficient psychological stimulation, frequent infections, and inadequate nutrition, remains a critical issue. Although economic growth alone cannot fully address the prevalence of stunting, there exists a robust correlation between a country's income level and childhood stunting rates. Countries with higher incomes tend to have lower rates of childhood stunting. Notably, while childhood stunting is declining worldwide, it remains persistent in Africa. Consequently, this study aims to assess the prevalence of childhood stunting and its determinants in low- and lower-middle-income African countries

## Method

This study conducted a secondary analysis of standard demographic and health surveys in low- and lower-middle-income African countries spanning the period from 2010 to 2022. The analysis included a total sample of 204,214 weighted children under the age of five years. To identify the determinants of stunting, we employed a multilevel mixed-effect model, considering the three levels of variables. The measures of association (fixed effect) were determined using the adjusted odds ratio at a 95% confidence interval. Significance was declared when the association between the outcome variable and the explanatory variable had a p-value less than 0.05.

www.dhsprogram.com/data/available-datasets.cfm].

**Funding:** The author(s) received no specific funding for this work.

**Competing interests:** The authors have declared that no competing interests exist.

## Result

In low and lower-middle-income African countries, 31.28% of children under five years old experience stunting, with a 95% confidence interval ranging from 31.08% to 31.48%. The results from a multilevel mixed-effect analysis revealed that 24 months or more of age of child, male gender, low and high birth weight, low and high maternal BMI, no and low maternal education, low household wealth index, multiple (twin or triplet) births, rural residence, and low income of countries were significantly associated with childhood stunting.

## Conclusion

Stunting among children under five years of age in low- and lower-middle-income African countries was relatively high. Individual, community, and country-level factors were statistically associated with childhood stunting. Equally importantly, with child, maternal, and community factors of stunting, the income of countries needs to be considered in providing nutritional interventions to mitigate childhood stunting in Africa.

## Introduction

Undernutrition is a major global public health concern affecting children under the age of five, since it delays their cognitive and physical development and raises their risk of disease and mortality [1]. Stunting is impaired growth and development that children experience due to inadequate psychological stimulation, frequent infections, and poor nutrition [2]. A child is deemed stunted if his or her height for age is more than two standard deviations below the WHO Child Growth Standards median [2]. The prevalence of stunting among children under 5 years of age is defined as the percentage of children whose height-for-age is more than two standard deviations below the median of the World Health Organization's (WHO) Child Growth Standards [3].

In 2020, there were 149.2 million stunted children under the age of five worldwide [4]. Stunting has decreased, from 40% of children under five worldwide in 1990 to 22% (144 million children) in 2021 [4]. It accounts for 45% of mortality in children under the age of five in low and middle income countries including in Africa [5]. According to the WHO, the current prevalence of stunting among children under five years of age in Africa is estimated to be 30.80% [3].

The Sustainable Development Goal (SDG) 2.2 aimed to end all forms of malnutrition, including stunting and wasting in children under 5 years of age, and address the nutritional needs of adolescent girls, pregnant and lactating women, and older persons by 2030 [6]. However, it appears the target will be hardly attained in economically disadvantaged countries in Africa unless a bunch of strategies and interventions are implemented [4]. Various policies and strategies have been implemented in Africa to address the SDGs. Notably, the World Health Organization (WHO) member states have adopted a strategic plan to reduce malnutrition in Africa [7–9]. This plan prioritizes several key interventions, including reinforcing legislation and food safety standards, using fiscal measures to promote healthy food choices, and integrating essential nutrition actions into health service delivery platforms. Additionally, the Africa Regional Nutrition Strategy 2015–2025, developed by the African Union (AU), underscores the significance of combating chronic undernutrition (stunting) as a primary objective for nutrition interventions [10–12]. The strategy acknowledges the crucial connection between

childhood stunting and the increasing prevalence of obesity, hypertension, and other non-communicable diseases across many African nations. It advocates evidence-based interventions and cost-effective development approaches.

As per studies [13–17], male gender, child age, multiple births, low birth weight, low maternal educational level, low maternal body mass index, poor household wealth, high illiteracy rate, maternal occupation, household income, antenatal care service utilization, and source of water were among the individual and community factors registered to have contributed to childhood stunting. This study included levels of factors at the country level, considering inter-country variation in stunting.

Though economic growth alone is not enough to reduce the prevalence of stunting, there is a strong association between the income of a country and childhood stunting [18, 19]. Countries with higher incomes have lower rates of childhood stunting [18, 19]. There is a decline in the number of stunted children throughout the globe but in Africa [4]. Hence, this study aimed to assess the prevalence of childhood stunting and its determinants in low and lower-middle income African countries.

## Method

### Data source, setting and sampling

This study conducted a secondary analysis of standard demographic and health surveys in low and lower-middle-income African countries spanning the period from 2010 to 2022. The study used the children's recode datasets, which were accessed from the Monitoring and Evaluation to Assess and Use Results Demographic and Health Survey (MEASURE DHS) program and are available in the public domain at https://www.dhsprogram.com. The MEASURE DHS provides researchers with data for formal requests for registration and submission of the proposed projects. A total of 32 low- and lower-middle-income African countries were selected for the study based on the availability of the DHS between 2010 and 2022 and the availability of WHO height for the age of the child Growth Standard score in their DHS data. The DHS data are collected using two-stage stratified sampling. The first stage is the selection of enumeration areas, also called sampling clusters, and the second stage is the selection of a sample of households in the clusters, where actual data are collected. Hence, DHS data are hierarchical (at two levels). A total sample of 204,214 weighted children under the age of five years was included in the analysis of this study.

### Variables

The outcome variable of the study was stunting among children under five years of age. Stunting was categorized using a binary code: 1 for stunted children and 0 for non-stunted children. Children were considered stunted if their height-for-age measurement fell more than two standard deviations below the WHO Child Growth Standards median [2].

The selection of explanatory variables for this study was guided by established principles and scholarly references [20]. Specifically, we conducted an extensive literature review, developed a theoretical framework, and considered data availability when choosing these variables [13–17, 20].

The explanatory variables of this study were grouped into three levels. At the first level, individual-level variables such as child age, gender, birth weight, maternal body mass index (BMI), maternal age, maternal education, maternal occupation, household wealth index, antenatal care (ANC) visit, place of delivery, media exposure, type of gestation, and source of water were included. At the second level, community-level variables such as distance to health facilities, residence, community women's literacy, community media utilization, and community

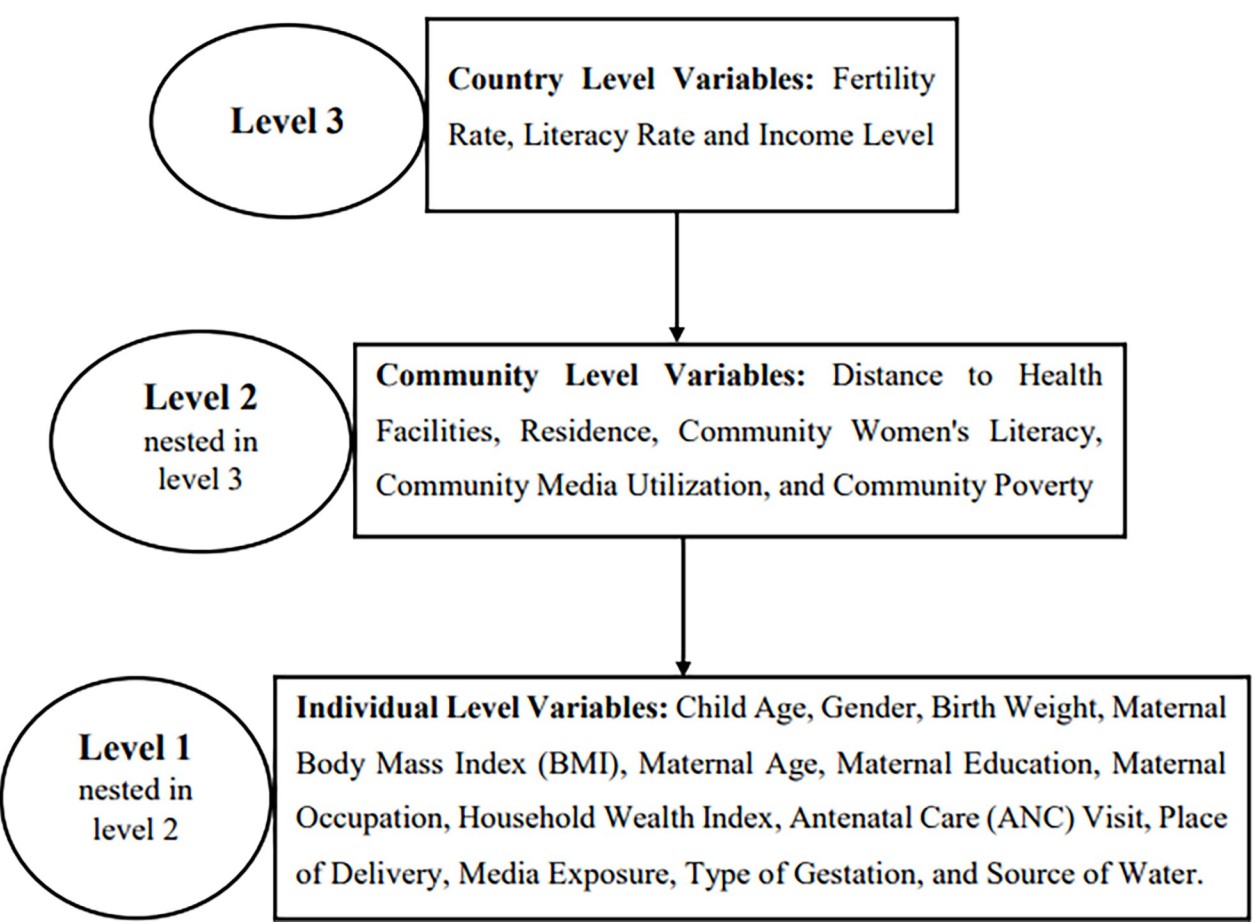

**Fig 1. Level of explanatory variables of the study.**

poverty were included. At the third level, country-level variables such as fertility rate, literacy rate, and income level were incorporated into the study (Fig 1). So the DHS data used for this study were of different years; the year of the surveys was used as a control variable to avoid the differential time effect.

## Data cleaning and management

The archive of children's recode data in Stata format for all countries was extracted from the Measure DHS website, https://www.dhsprogram.com. The data cleaning process of this study involved several key steps. First, we removed irrelevant data by identifying and excluding any variables or observations unrelated to stunting, maternal health, nutrition, and other relevant factors. Next, we checked for duplicate records within the dataset, eliminating any repeated observations to prevent bias in our analysis. Third, we followed the DHS guide to handle missing data. Fourth, we ensured consistency by inspecting variable names and maintaining uniformity across the dataset. With clean data, we proceeded to describe frequencies and rates using tables and charts. Finally, we performed statistical modeling, exploring associations between stunting and associated factors. These steps were essential to ensuring the reliability and validity of subsequent analyses. Weighted frequencies and percentages were presented in tables and charts.

## The regression model

Considering the three levels of variables used, this study applied a multilevel mixed effect model to identify determinants of stunting among children under the age of five years. Accordingly, five mixed-effect models were fitted. First, the null model (model 0), a model with outcome and no explanatory variables; second, model I, a model with an outcome and individual-level explanatory variables; third, model II, a model with an outcome and community-level explanatory variables; fourth, model III, a model with an outcome and country-level explanatory variables; and finally, model IV, a model with an outcome variable and all three levels of variables.

The equation for a three-level multilevel mixed effect regression, a statistical model that accounts for the hierarchical structure of data with three levels, can be written as [21, 22]:

$$y_{ijk} = \beta_{0k} + \beta_{1k}x_{ijk} + r_{ijk} + u_{ijk} + e_{ijk}$$

Where $y_{ijk}$ is the dependent variable for the $i^{th}$ individual in the $j^{th}$ community in the $k^{th}$ country, $\beta_{0k}$ and $\beta_{1k}$ are the intercept and slope coefficients for the $k^{th}$ country, $x_{ijk}$ is the independent variable for the $i^{th}$ individual in the $j^{th}$ community in the $k^{th}$ country, $r_{ijk}$ is the random effect for the $j^{th}$ community in the $k^{th}$ country, $u_{ijk}$ is the random effect for the $i^{th}$ individual in the $j^{th}$ community in the $k^{th}$ country and $e_{ijk}$ is the residual error for the $i^{th}$ individual in the $j^{th}$ community in the $k^{th}$ country.

In our nested model with three levels, we operate under the following assumptions: Variations at different levels are intricately linked to the levels above them. At the foundation, Level 1, individual observations or measurements—variations in stunting among children—serve as the fundamental building blocks for our entire model. As we ascend to Level 2 (communities), we encounter community-level variations influenced by correlations between individuals within the same community. These community-level dynamics, in turn, reverberate upward to impact the highest level (Level 3), which represents countries. Importantly, our model explicitly accounts for the hierarchical structure inherent in the data: individuals nest within communities, and communities nest within countries.

The random effect and fixed effect components of the mixed effect model were assessed. The parameters variance, intra-class correlation coefficient, and median odds ratio were used to assess measures of variation (random effect). The MOR, and ICC were determined by the following equations [23]; $MOR = e^{0.95\sqrt{VC}}$ and $ICC = \frac{VC}{VC+3.29} \times 100\%$, where; VC = variance of the cluster for respective model.

The measures of association (fixed effect) were determined using the adjusted odds ratio at a 95% confidence interval. The significance of the association between the outcome variable and the explanatory variable was declared at a p-value less than the level of significance (0.05). By considering the nested nature of multilevel mixed effects, model comparisons were made using deviance (-2LL).

## Ethical consideration and approval

This study was based on analysis of existing survey datasets in the public domain that are freely available online with all the identifier information anonymized, no ethical approval was required. The first author was obtained authorization for the download and usage of the DHS dataset of all countries included in the analysis from MEASURE DHS.

## Results

A total of 204,214 (50.54% males and 49.46% females) weighted children under the age of five years were included in the analysis of the study. More than half (57.27%) of children were aged 24–59 months. About half (51.67%) of the subjects were born low-birth-weighted. Importantly, 62.06% of the subjects were born to mothers with a normal body mass index. The majority (71.04%) of the subjects were born to mothers aged 20–34 years. More than two-thirds (69.24%) of children were residents of rural areas, and a bit more than half (51.56%) of the subjects were from low-income countries (Table 1).

Out of 103,216 male children, one-third (33.30%) were stunted. By place of residence, 34.80% of rural and 22.89% of urban resident children were stunted. In addition, 35.52% of children with low income and 26.48% with lower-middle income were stunted (Table 1).

### Prevalence of childhood stunting in low and lower-middle income African countries

The pooled prevalence of childhood stunting in low and lower-middle-income African countries was found to be 31.28% at 95% CI (31.08, 31.48). The prevalence was higher (54.51%) in Burundi and lower (17.92%) in Kenya (Fig 2).

### Random effect and model comparison

The value of variance in the null model of the random effect shows that there was variation in childhood stunting among countries (τ = 0.51, p<0.001) and communities (τ = 0.58, p<0.001). About 13% and 15% of the total variation in childhood stunting was attributed to variation across countries (13.41%) and across communities (ICC = 14.99%), respectively. The variation of stunting among countries and communities (clusters) remains significant even after all levels of variables were fitted to it. The unexplained heterogeneity (MOR in the null model) in stunting among countries and communities was 1.97 and 2.06, respectively, without fitting any explanatory variables. The heterogeneity was reduced to 1.51 among countries and 1.54 among communities after all levels of the explanatory variables were fitted (Table 2).

Regarding the model comparison, model IV, the model with a large log likelihood and small deviance values was selected as the best fit (Table 2).

### Fixed effect of childhood stunting in low and lower-middle income African countries

The multilevel mixed effect analysis of this study revealed that age of child, gender, birth weight, maternal BMI, maternal education, wealth index, type of gestation, place of residence, and income level of countries were significantly associated with childhood stunting.

At the individual level, At the individual level, the odds of stunting were 1.76 times as high among children aged 24–59 months (AOR = 1.76 at 95% CI: 1.70, 1.83) compared to children aged less than 24 months. The odds of stunting were 1.37 times as high among male children (AOR = 1.37 at 95% CI: 1.32, 1.42) compared to females. On the one hand, the odds of stunting were 1.38 times as high among children with low birth weight (AOR = 1.38 at 95% CI: 1.31, 1.43); on the other hand, the odds of stunting were reduced by 33% for children with high birth weight (AOR = 0.77 at 95% CI: 0.72, 0.83) when compared to children with normal birth weight. Similarly, the odds of stunting were 1.26 times as high among children born to mothers with low BMI (AOR = 1.26 at 95% CI: 1.19, 1.33) and reduced by 30% for children born to mothers with high BMI (AOR = 0.70 at 95% CI: 0.67, 0.73), compared to children born to mothers with normal BMI. The odds of stunting were 2.30, 2.08, and 1.68 times as high among

**Table 1. Description of childhood stunting by level of variables in low and lower-middle income African countries (N = 204,214).**

| Variables at respective levels | | Frequency (Percent) | Childhood stunting | |
|---|---|---|---|---|
| | | N (%) | Stunted [N (%)] | Not stunted [N (%)] |
| **Individual level variables** | | | | |
| Child age | 0–23 months | 60,743 (42.73) | 15,789 (25.99) | 44,954 (74.01) |
| | 24–59 months | 81,417 (57.27) | 27,592 (33.89) | 53,825 (66.11) |
| Gender | Male | 103,216 (50.54) | 34,366 (33.30) | 68,850 (66.70) |
| | Female | 100,998 (49.46) | 29,225 (28.94) | 71,773 (71.06) |
| Birth weight | Low | 97,110 (51.67) | 36,406 (37.49) | 60,705 (62.51) |
| | Normal | 77,551 (41.26) | 20,733 (26.73) | 56,818 (73.27) |
| | High | 13,273 (7.06) | 3,277 (24.69) | 9,996 (75.31) |
| Maternal BMI | Low | 15,855 (9.48) | 6,592 ( 41.58) | 9,263 (58.42) |
| | Normal | 103,781 (62.06) | 36,142 (34.83) | 67,639 (65.17) |
| | High | 47,602 (28.46) | 10,255 (21.54) | 37,347 (78.46) |
| Maternal age | 15–19 | 11,499 (5.63) | 3,699 (32.17) | 7,800 (67.83) |
| | 20–34 | 145,079 (71.04) | 44,891 (30.94) | 100,188 (69.06) |
| | 35–49 | 47,635 (23.33) | 15,001 (31.49) | 32,635 (68.51) |
| Maternal educational status | No education | 80,533 (39.44) | 30,006 (37.26) | 50,527 (62.74) |
| | Primary | 61,869 (30.30) | 20,574 (33.25) | 41,295 (66.75) |
| | Secondary | 52,142 (25.53) | 11,809 (22.65) | 40,334 (77.35) |
| | Higher | 9,669 (4.73) | 1,203 (12.44) | 8,466 (87.56) |
| Maternal occupation | Not working | 92,225 (45.23) | 27,609 (29.94) | 64,615 (70.06) |
| | Working | 111,688 (54.77) | 35,863 (32.11) | 75,825 (67.89) |
| Wealth index | Poorest | 45,457 (22.26) | 17,516 (38.53) | 27,940 (61.47) |
| | Poorer | 43,168 (21.14) | 15,540 (36.00) | 27,628 (64.00) |
| | Middle | 41,735 (20.44) | 13,104 (31.40) | 28,631 (68.60) |
| | Richer | 39,734 (19.46) | 11,038 (27.78) | 28,695 (72.22) |
| | Richest | 34,120 (16.71) | 6,393 (18.74) | 27,727 (81.26) |
| ANC visits | No visits | 16,902 (11.92) | 6,812 (40.31) | 10,089 (59.69) |
| | 1–3 visits | 42,679 (30.1) | 13,823 (32.39) | 28,856 (67.61) |
| | 4 or more | 82,226 (57.98) | 20,825 (25.33) | 61,401 (74.67) |
| Place of delivery | Home | 65,186 (33.85) | 26,051 (39.96) | 39,135 (60.04) |
| | Health facility | 127,370 (66.15) | 35,242 (27.67) | 92,128 (72.33) |
| Media exposure | Yes | 129,174 (63.38) | 34188 (26.47) | 94986 (73.53) |
| | No | 74,624 (36.62) | 29235 (39.18) | 45389 (60.82) |
| Type of gestation | Singleton | 197793 (96.86) | 60714 (30.70) | 137079 (69.30) |
| | Multiple | 6420 (3.14) | 2877 (44.81) | 3543 (55.19) |
| Source of water | Improved | 68381 (33.49) | 16975 (24.82) | 51407 (75.18) |
| | Unimproved | 135812 (66.51) | 46,611 (34.32) | 89,201 (65.68) |
| **Community level variables** | | | | |
| Distance to health facility | Not big problem | 111,194 (62.65) | 31,903 (28.69) | 79,291 (71.31) |
| | Big problem | 66293 (37.35) | 22754 (34.32) | 43539 (65.68) |
| Residence | Urban | 62,814 (30.76) | 14378 (22.89) | 48436 (77.11) |
| | Rural | 141,400 (69.24) | 49,213 (34.80) | 92,187 (65.20) |
| Community women literacy | Low | 120,191 (58.86) | 40389 (33.60) | 79802 (66.40) |
| | High | 84,023 (41.14) | 23,202 (27.61) | 60,821 (72.39) |
| Community media utilization | Low | 112,913 (55.29) | 38,181 (33.81) | 74,732 (66.19) |
| | High | 91,300 (44.71) | 25,410 (27.83) | 65,890 (72.17) |

(*Continued*)

**Table 1.** (Continued)

| Variables at respective levels | | Frequency (Percent) | Childhood stunting | |
|---|---|---|---|---|
| | | N (%) | Stunted [N (%)] | Not stunted [N (%)] |
| Community poverty | Low | 98220 (48.15) | 29053 (29.58) | 69167 (70.42) |
| | High | 105,768 (51.85) | 34488 (32.61) | 71280 (67.39) |
| **Country level variables** | | | | |
| Fertility rate | Low | 136223 (66.71) | 39413 (28.93) | 96810 (71.07) |
| | high | 67990 (33.29) | 24,178 (35.56) | 43,812 (64.44) |
| Literacy rate | Low | 98283 (48.13) | 31639 (32.19) | 66644 (67.81) |
| | High | 105,930 (51.87) | 31,952 (30.16) | 73,978 (69.84) |
| Income | Low | 105286 (51.56) | 37398 (35.52) | 67888 (64.48) |
| | Lower middle | 98928 (48.44) | 26193 (26.48) | 72734 (73.52) |

ANC: Antenatal care, BMI: Body mass index

children born to mothers with no formal education (AOR = 2.30 at 95% CI: 2.03, 2.60), primary schooling (AOR = 2.08 at 95% CI: 1.84, 2.35), and secondary schooling (AOR = 1.68 at 95% CI: 1.49, 1.90) as compared to children born to mothers with a higher level of education. Compared to children of households with the richest wealth index, the odds of stunting were 1.65, 1.64, 1.44, and 1.29 times as high for children of households with the poorest (AOR = 1.65 at 95% CI: 1.53, 1.77), poorer (AOR = 1.64 at 95% CI: 1.52, 1.76), middle (AOR = 1.44 at 95% CI: 1.34, 1.55), and richer (AOR = 1.29 at 95% CI: 1.21, 1.38) wealth indexes. The odds of stunting were 2.60 (AOR = 2.60 at 95% CI: 2.30, 2.93) times higher among children born multiple compared to children born singleton.

At the community level, the odds of stunting were increased by 10% among rural resident children (AOR = 1.10 at 95% CI: 1.04, 1.15) compared to urban resident children.

At the country level, the odds of childhood stunting were increased by 5% in low-income African countries (AOR = 1.05 at 95% CI: 1.01, 1.09) compared to lower-middle-income African countries (Table 3).

## Discussion

Although economic growth alone is not enough to reduce the prevalence of stunting, there is a strong association between the income of a country and childhood stunting [18, 19]. Countries with higher incomes have lower rates of childhood stunting [18, 19]. There is a decline in the number of stunted children throughout the globe but in Africa [4]. Thus, this study revealed the prevalence of childhood stunting and its determinants in low and lower-middle income African countries.

The prevalence of stunting among children under five years of age in low and lower-middle-income African countries was 31.28% at a 95% CI (31.08, 31.48). The prevalence of stunting in this study was higher than the WHO estimated prevalence of stunting among children under five years of age in Africa, which is 30.80% [3]. The higher prevalence of stunting in this study than the WHO estimate could be due to the fact that this study included only low- and lower-middle-income countries, while the WHO estimated stunting in Africa regardless of the level of income of the countries (in low-, lower-middle-, upper-middle-, and high-income countries). In addition, the confluence of instability, climate-related challenges, and economic hardships in various regions of Africa significantly contributes to the high prevalence of stunting among children in low and lower-middle income African countries [8, 24]. These intertwined factors collectively impact child growth and development, creating a formidable

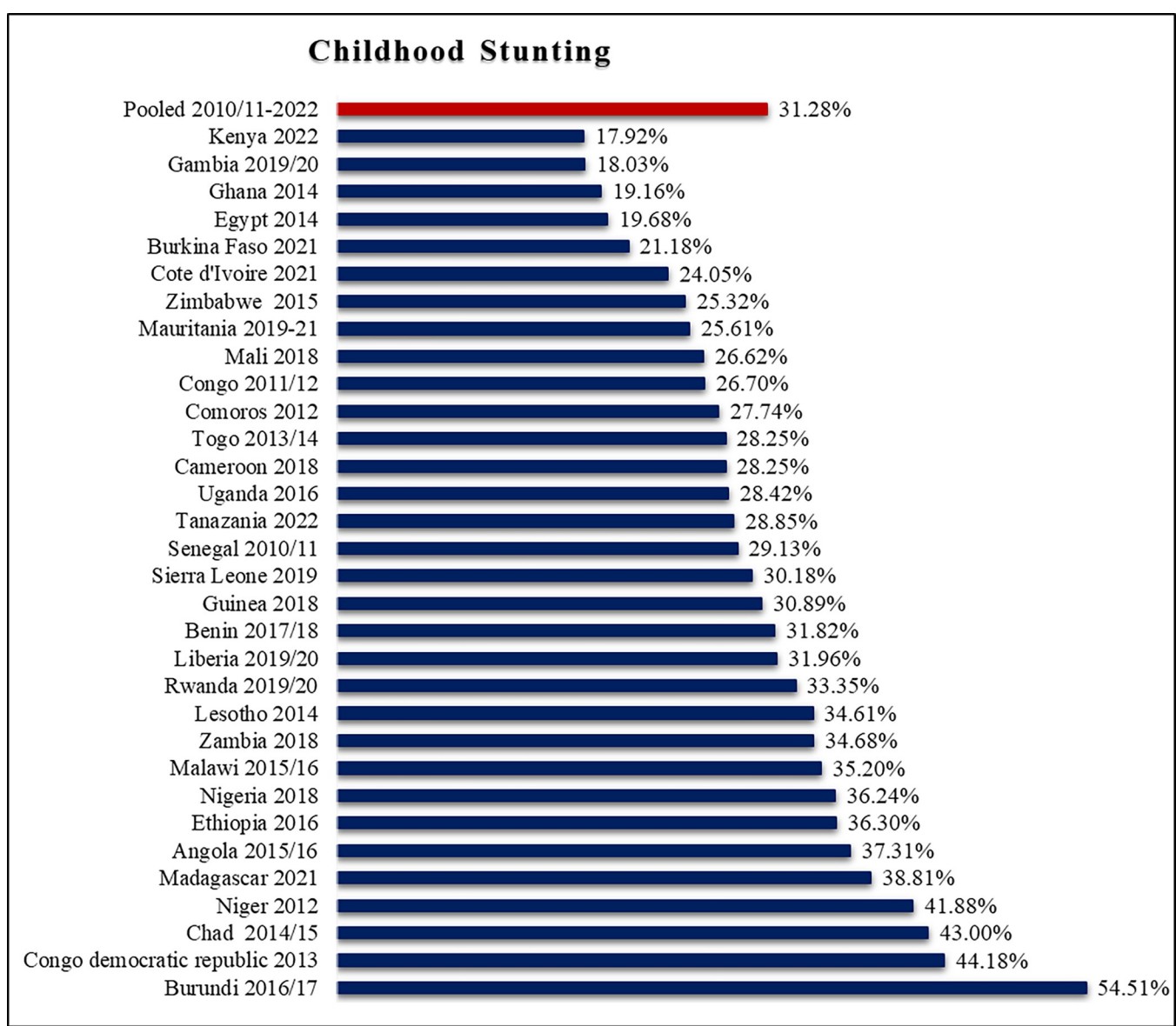

**Fig 2. The prevalence of childhood stunting in low and lower-middle income African countries.**

barrier to optimal health and well-being [8]. Thus, childhood stunting in low and lower-middle-income African countries is high, and the interventions that can mitigate the effects of stunting during pregnancy and early childhood are found to be critical.

Consistent with previous studies [25–27], the age of a child was significantly associated with stunting. The odds of stunting were high among babies aged 24 or more months compared to children aged less than 24 months. The reason behind the higher odds of stunting among older children than younger babies could be related to the longstanding feature of stunting. Stunting is a long-term condition that develops over a long period of time, and it is hard to reverse once it has happened [28]. Consequently, stunting is more common in older children who have experienced malnutrition for a longer period of time than in younger infants who have not [28].

**Table 2. Random effect and model comparison of childhood stunting in low and lower-middle income African countries.**

| Parameters | | Null Model | Model I | Model II | Model III | Model IV |
|---|---|---|---|---|---|---|
| **Random effect** | | | | | | |
| Variance (τ) | Country | 0.51 ** | 0.20 ** | 0.32** | 0.41** | 0.19 * |
| | Community | 0.58 ** | 0.22 ** | 0.38 ** | 0.35** | 0.21 * |
| ICC (%) | Country | 13.42 | 5.73 | 8.86 | 11.08 | 5.45 |
| | Community | 14.99 | 6.27 | 10.35 | 9.61 | 6.00 |
| MOR | Country | 1.97 | 1.53 | 1.71 | 1.84 | 1.51 |
| | Community | 2.06 | 1.56 | 1.80 | 1.75 | 1.54 |
| **Model comparison** | | | | | | |
| Log likelihood | | -127704.78 | -41044.25 | -108122.77 | -126199.46 | -40905.41 |
| Deviance | | 255,409.56 | 82,088.50 | 216,245.54 | 252,398.92 | 81,810.82 |

ICC: intra-class correlation coefficient, MOR: Median Odds Ratio

**: p<0.001

*: p<0.05.

This study found that the odds of stunting were higher among male children than females. Surprisingly, several previous studies have also witnessed the same [27, 29–31]. The reason behind gender-based inequalities in childhood undernutrition is not well understood. However, as a consequence of their higher birth weight than females, male children require more calories, which increases the risk that they will have undernutrition [32, 33]. Moreover, it has been suggested that male children are more prone to the three nutrient deficiencies (stunting, wasting, and underweight) due to their increased hunger than female children; therefore, breastfeeding alone may not be adequate for them [34, 35].

The abnormal birth weight was a significant determinant of childhood stunting. On the one hand, the odds of stunting were higher among children with low birth weight and on the other hand, the odds of stunting were lower among children with high birth weight. This was in agreement with previous findings [36–38]. Low birth weight is a common outcome of pregnancy-related malnutrition, which can impede the fetus's growth and development [39]. This can lead to a child being born with a smaller size and lower weight than normal, which raises their risk of stunting later in life [37, 39]. High-birth-weight babies are more likely to be over-nourished than undernourished, and it has been shown that high birth weight is progressive throughout childhood and is a possible cause of obesity in adulthood [40]. These findings show that certain characteristics of childhood undernutrition and subsequent linear development have antecedents in the prenatal period, and they emphasize the significance of maternal nutrition both preconceptionally and antenatally [39, 41–43].

According to this study, low and high maternal BMI were significantly associated with high and low odds of childhood stunting, respectively. This similar finding was also evidenced by previous studies [44, 45]. The reason for the consistent association between maternal BMI and child anthropometric failures may be the intrauterine intergenerational transmission of low maternal BMI during pregnancy, which puts infants at risk of low birth weight and small stature for gestational age and forms the fetal origins of subsequent childhood undernutrition [43, 46].

In line with previous scientific evidences [47–49], maternal education was significantly associated with childhood stunting. Accordingly, the odds of stunting were higher among children born to mothers with no formal education, primary schooling and secondary schooling as compared to children born to mothers with higher level of education. The association of

**Table 3.  Fixed effect of childhood stunting in low and lower-middle income African countries.**

| Factors at respective levels | | Model I | Model II | Model III | Model IV |
|---|---|---|---|---|---|
| | | AOR (95% CI) | AOR (95% CI) | AOR (95% CI) | AOR (95% CI) |
| Survey year | 2010–2014 | 1 | 1 | | 1 |
| | 2015–2019 | 1.18 (1.10, 1.25) | 1.07 (1.04, 1.09) | 1.11 (1.09, 1.14) | 1.08 (0.98, 1.16) |
| | 2020–2022 | 0.86 (0.80, 0.92) | 0.72 (0.70, 0.74) | 0.72 (0.70, 0.74) | 0.76 (0.71, 0.82) |
| Child age | 0–23 months | 1 | | | 1 |
| | 24–59 months | 1.79 (1.73, 1.85) | | | 1.76 (1.70, 1.83)* |
| Gender | Male | 1.36 (1.32, 1.41) | | | 1.37 (1.32, 1.42)* |
| | Female | 1 | | | 1 |
| Birth weight | Low | 1.30 (1.24, 1.36) | | | 1.38 (1.31, 1.43)* |
| | Normal | 1 | | | 1 |
| | High | 0.78 (0.73, 0.84) | | | 0.77 (0.72, 0.83)* |
| Maternal BMI | Low | 1.27 (1.21, 1.35) | | | 1.26 (1.19, 1.33)* |
| | Normal | 1 | | | 1 |
| | High | 0.67 (0.64, 0.70) | | | 0.70 (0.67, 0.73)* |
| Maternal age | 15–19 | 1.09 (1.02, 1.16) | | | 1.05 (0.93, 1.17) |
| | 20–34 | 1 | | | 1 |
| | 35–49 | 0.99 (0.95, 1.03) | | | 0.99 (0.94, 1.04) |
| Maternal educational status | No education | 2.15 (1.90, 2.43) | | | 2.30 (2.03, 2.60)* |
| | Primary | 2.20 (1.94, 2.48) | | | 2.08 (1.84, 2.35)* |
| | Secondary | 1.71 (1.51, 1.93) | | | 1.68 (1.49, 1.90)* |
| | Higher | 1 | | | 1 |
| Maternal occupation | Not working | 1 | | | 1 |
| | Working | 1.16 (0.91, 1.20) | | | 1.14 (0.90, 1.18) |
| Wealth index | Poorest | 1.70 (1.59, 1.82) | | | 1.65 (1.53, 1.77)* |
| | Poorer | 1.69 (1.58, 1.81) | | | 1.64 (1.52, 1.76)* |
| | Middle | 1.47 (1.38, 1.57) | | | 1.44 (1.34, 1.55)* |
| | Richer | 1.30 (1.22, 1.39) | | | 1.29 (1.21, 1.38)* |
| | Richest | 1 | | | 1 |
| ANC visits | No visits | 1.01 (0.81, 1.25) | | | 1.02 (0.79, 1.23) |
| | 1–3 visits | 1.03 (0.93, 1.18) | | | 1.00 (0.92, 1.16) |
| | 4 or more | 1 | | | 1 |
| Place of delivery | Home | 1.01 (0.97, 1.06) | | | 1.00 (0.96, 1.05) |
| | Health facility | 1 | | | 1 |
| Media exposure | Yes | 1 | | | 1 |
| | No | 1.12 (0.82, 1.17) | | | 1.09 (0.79, 1.13) |
| Type of gestation | Singleton | 1 | | | 1 |
| | Multiple | 2.56 (2.28, 2.89) | | | 2.60 (2.30, 2.93)* |
| Source of water | Improved | 1 | | | 1 |
| | Unimproved | 0.99 (0.95, 1.03) | | | 0.97 (0.93, 1.01) |
| Distance to health facility | Not big problem | | 1 | | 1 |
| | Big problem | | 1.14 (1.11, 1.16) | | 0.95 (0.92, 1.02) |
| Residence | Urban | | 1 | | 1 |
| | Rural | | 1.66 (1.62, 1.70) | | 1.10 (1.04, 1.15)* |
| Community women literacy | Low | | 1.24 (1.18, 1.31) | | 1.03 (0.98, 1.07) |
| | High | | 1 | | 1 |
| Community media utilization | Low | | 1.36 (1.29, 1.43) | | 1.02 (0.98, 1.07) |
| | High | | 1 | | 1 |

*(Continued)*

**Table 3.** (Continued)

| Factors at respective levels | | Model I | Model II | Model III | Model IV |
|---|---|---|---|---|---|
| | | AOR (95% CI) | AOR (95% CI) | AOR (95% CI) | AOR (95% CI) |
| Community poverty | Low | | 1 | | 1 |
| | High | | 0.94 (0.90, 1.02) | | 0.96 (0.92, 1.04) |
| Fertility rate | Low | | | 1.26 (1.23, 1.28) | 0.99 (0.95, 1.04) |
| | high | | | | |
| Literacy rate | Low | | | 0.99 (0.97, 1.01) | 0.85 (0.72, 1.08) |
| | High | 1 | | 1 | 1 |
| Income | Low | | | 1.39 (1.36, 1.42) | 1.05 (1.01, 1.09)* |
| | Lower middle | | | 1 | 1 |

ANC: Antenatal Care, AOR: Adjusted Odds Ration, BMI: Body mass index, CI: Confidence Interval

Interval, *: anificant (p-value<0.05).

maternal education with childhood stunting was attributable to the health and growth benefits that accompany increased resources and occupational standing among educated mothers [48]. In addition, maternal education improves the ability of the mother to foster a healthy environment for the child's intrauterine growth [48].

Importantly, the odds of stunting increased with a decrease in the household wealth quintile. Compared to children of households with the richest wealth index, the odds of stunting were higher for children of households with the richer, middle, poorer, and poorest wealth indexes. This finding was in agreement with existing evidences [50–52]. It is plausible that children from low-income households are more likely to have growth failure due to poor nutrition, a higher risk of illness, and difficulties accessing basic health services [53].

Regarding the type of gestation, the odds of stunting were higher among children born multiple (twin or triplet) compared to children born singleton. This finding was consistent with previous evidence [52, 54]. This could be due to the fact that low birth weight and competition for nutrition, which is common among twins or triplets, lead to stunting later in childhood [55].

When compared to children in urban areas, the odds of stunting were high for children in rural areas. This was in agreement with previous findings [56–58]. Access to appropriate health care is challenging for people living in rural areas, and they lack immediate interventions for acute undernutrition [59] among their babies, which can progress to chronic growth failure (stunting). Poor household expenditure, unhealthy snacks, and poor sanitation, which are more common in rural areas, can contribute to stunting among children of rural residents compared to urban ones [56].

Equally importantly, this study found that low income level of countries was significantly associated with higher odds of childhood stunting compared to lower-middle income countries. This was in line with finding from another study [18]. The low income level of a country can lead to stunting among children due to a lack of resources and access to basic needs such as food, water, sanitation, and healthcare [19]. Investing in childhood nutrition and providing access to basic needs can help reduce stunting and expand the economic opportunities of children [2, 19].

The findings of this study should be regarded with the following strengths and limitations: The use of nationally representative DHS data and the application of an appropriate advanced model were virtues of this study. However, due to the cross-sectional nature of the surveys, the associations between independent variables and the outcome ascertained by this study cannot show causal associations.

## Conclusion

Stunting among children under five years of age in low- and lower-middle-income African countries was relatively high. Individual, community, and country-level factors were statistically associated with childhood stunting. Equally importantly, with child, maternal, and community factors of stunting, the income of countries needs to be considered in providing nutritional interventions to mitigate childhood stunting in Africa.

## Supporting information

**S1 Checklist. Human participants research checklist.**
(DOCX)

## Author Contributions

**Conceptualization:** Soliyana Adisu Gezhegn, Dejen Tegegne Dagnew, Abebe Tilahun Mekonenne, Genetu Tadese Aweke.

**Data curation:** Tadesse Tarik Tamir, Soliyana Adisu Gezhegn, Dejen Tegegne Dagnew, Abebe Tilahun Mekonenne, Genetu Tadese Aweke.

**Formal analysis:** Tadesse Tarik Tamir.

**Investigation:** Tadesse Tarik Tamir, Abebe Tilahun Mekonenne, Genetu Tadese Aweke, Ayenew Molla Lakew.

**Methodology:** Tadesse Tarik Tamir, Abebe Tilahun Mekonenne, Ayenew Molla Lakew.

**Software:** Tadesse Tarik Tamir, Dejen Tegegne Dagnew, Ayenew Molla Lakew.

**Supervision:** Ayenew Molla Lakew.

**Validation:** Genetu Tadese Aweke, Ayenew Molla Lakew.

**Visualization:** Soliyana Adisu Gezhegn, Dejen Tegegne Dagnew, Abebe Tilahun Mekonenne, Genetu Tadese Aweke.

**Writing – original draft:** Tadesse Tarik Tamir.

**Writing – review & editing:** Soliyana Adisu Gezhegn, Dejen Tegegne Dagnew, Abebe Tilahun Mekonenne, Ayenew Molla Lakew.

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
