## [Decision Letter · Decision Letter 0]

17 Mar 2024

PONE-D-24-04373Prevalence of Childhood Stunting and Determinants in Low and Lower-Middle Income African Countries: Evidence from Standard Demographic and Health SurveyPLOS ONE

Dear Dr. Tamir,

Thank you for submitting your manuscript to PLOS ONE. After careful consideration, we feel that it has merit but does not fully meet PLOS ONE’s publication criteria as it currently stands. Therefore, we invite you to submit a revised version of the manuscript that addresses the points raised during the review process.

We look forward to receiving your revised manuscript.

Kind regards,

Satyajit Kundu

Academic Editor

PLOS ONE

Additional Editor Comments:

The authors are requested to revise the manuscript carefully based on the reviewer's comments to avoid the further delay.

Reviewers' comments:

Reviewer's Responses to Questions

**Comments to the Author**

1. Is the manuscript technically sound, and do the data support the conclusions?

Reviewer #1: Yes

Reviewer #2: Yes

Reviewer #3: Yes

2. Has the statistical analysis been performed appropriately and rigorously? 

Reviewer #1: Yes

Reviewer #2: Yes

Reviewer #3: No

3. Have the authors made all data underlying the findings in their manuscript fully available?

Reviewer #1: Yes

Reviewer #2: Yes

Reviewer #3: Yes

4. Is the manuscript presented in an intelligible fashion and written in standard English?

Reviewer #1: Yes

Reviewer #2: Yes

Reviewer #3: Yes

5. Review Comments to the Author

Reviewer #1: This study aimed to assess the prevalence of childhood stunting and its determinants in low-and lower-middle-income African countries. Studying stunting in developing countries is very important and I encourage authors to continue their good work. I have a few suggestions, however, to help improve the manuscript.

1. In the introduction, I will suggest the authors add some policies, strategies and interventions done in Africa to meet the SDG 2.2 by 2030 for stunting and undernutrition.

2. In the variables section, i will suggest the authors clearly state how stunting was operationally defined in the DHS dataset and the responses and not just give how coding was done.

3. The authors should please provide references for what informed the explanatory variables.

4. In the discussion section, I will suggest the authors give reasons with references for the high prevalence of stunting in low-and lower-middle-income African countries and not just compare with the WHO findings.

Reviewer #2: The authors have done a comprehensive analysis of available standard demographic datasets to investigate the prevalence of childhood stunting throughout Africa and and its determinants in African countries. I think the findings will be really helpful to adopt continent-wide policy making by governments and international agencies. I hope this article will add value to the public health related literatures on low and middle income African countries.

Reviewer #3: I would like to thank the editor for giving me opportunity to review this article. Here are few points I am concerned about.

Methodology

1. The wording of second line in section data source, setting and sampling seems need correction.

2. Mentioning how researchers came with those potential predictors in section variables? Referencing again basically. I believe they are 7-11.

3. The description provided for the expression of the generic model presented in section The Regression Model need amendments? Basically, correcting subscripts.

4. In the regression model what assumptions were made? I see three random terms there. What distribution they have? Is r_jk independent of u_ik?

5. Since sample size is large will considering a 0.01 level make sense? So, 99% CI needs to be calculated instead of 95% CI.

6. Did authors do data cleaning? If so, what was their procedure?

Results

7. With table 2 researchers selected model IV. Is there a need for mentioning results from other models in table 3?

8. After considering 0.01 level, I believe there will more variables that will be insignificant. In that case will keeping only the significant variables in the model make sense?

9. If authors do not agree with 0.01 level, will keeping only significant variable in the model make sense?

10. “At the individual level, the odds of stunting were 1.76 times higher among children aged 24-59 months (AOR = 1.76 at 95% CI: 1.70, 1.83) compared to children aged less than 24 months.” In my opinion, this is not correct. The correct interpretation would be “At the individual level, the odds of stunting were 1.76 times as high among children aged 24-59 months (AOR = 1.76 at 95% CI: 1.70, 1.83) compared to children aged less than 24 months.” If authors agree, please change rest of the interpretations in this way.

6. PLOS authors have the option to publish the peer review history of their article (what does this mean?). If published, this will include your full peer review and any attached files.

Reviewer #1: No

Reviewer #2: No

Reviewer #3: No

---

## [Author Response · Author response to Decision Letter 0]

20 Mar 2024

Response to Comments

Manuscript ID: PONE-D-24-04373

Title: Prevalence of Childhood Stunting and Determinants in Low and Lower-Middle Income African Countries: Evidence from Standard Demographic and Health Survey

Journal: PLOS ONE

Subject: Submission of revised manuscript

I hope this letter finds you well. We appreciate the diligent efforts of the editorial team in facilitating the review process for our manuscript. Additionally, we extend our gratitude to the reviewers for their valuable time and thoughtful feedback, which significantly contributed to enhancing the quality of our work.

The constructive comments provided by the reviewers have been instrumental in refining our study. We are pleased to note that the reviewers share our assessment of the scientific significance of our findings. In response to their suggestions, we have meticulously addressed each point raised. Please find our comprehensive responses to the comments below.

Furthermore, we have attached the revised manuscript file separately for your convenience. We believe that the revisions strengthen the manuscript and align it more closely with the journal’s scope and standards.

Thank you for considering our work for publication. We look forward to your feedback and hope that our revised submission meets the high standards set by PLOS ONE.

Best regards,

Tadesse Tarik Tamir, corresponding author (on behalf of all authors)

University of Gondar, Gondar, Ethiopia

Response to editors and reviewers’ and comments

Reviewer #1: This study aimed to assess the prevalence of childhood stunting and its determinants in low-and lower-middle-income African countries. Studying stunting in developing countries is very important and I encourage authors to continue their good work. I have a few suggestions, however, to help improve the manuscript.

Response: Dear reviewer, we sincerely appreciate your enthusiasm for our manuscript’s subject and hypotheses. Additionally, we value your detailed perspectives and insightful comments. 

1. In the introduction, I will suggest the authors add some policies, strategies and interventions done in Africa to meet the SDG 2.2 by 2030 for stunting and undernutrition.

Response: Dear reviewer, thank you for your insightful suggestion. We incorporated policies, strategies, and interventions done in Africa for meeting SDG 2.2 by 2030. Kindly refer to the introduction section in our revised manuscript.

2. In the variables section, i will suggest the authors clearly state how stunting was operationally defined in the DHS dataset and the responses and not just give how coding was done.

Response: Dear reviewer, Thank you for your invaluable insights regarding the operational definition of the outcome variable. In response to your suggestion, we have clearly stated how the dependent variable was operationally defined. Kindly refer to the relevant section in our revised manuscript.

3. The authors should please provide references for what informed the explanatory variables.

Response: Dear reviewer, I sincerely appreciate your valuable insights regarding variable selection in our study. In response to your suggestion, we have included references that informed our decision-making process regarding explanatory variables. Please find these details in the variable section of our revised manuscript.

4. In the discussion section, I will suggest the authors give reasons with references for the high prevalence of stunting in low-and lower-middle-income African countries and not just compare with the WHO findings.

Response: Dear reviewer, I extend my gratitude for your insightful suggestion. In light of this, we have elucidated the reasons behind the high prevalence of stunting in low- and lower-middle-income African countries. Kindly find the point in our revised manuscript. 

Reviewer #2: The authors have done a comprehensive analysis of available standard demographic datasets to investigate the prevalence of childhood stunting throughout Africa and and its determinants in African countries. I think the findings will be really helpful to adopt continent-wide policy making by governments and international agencies. I hope this article will add value to the public health related literatures on low and middle income African countries.

Response: Dear reviewer, we sincerely appreciate your enthusiasm for our manuscript’s subject and hypotheses. Additionally, we value your detailed perspectives and insightful comments.

Reviewer #3: I would like to thank the editor for giving me opportunity to review this article. Here are few points I am concerned about.

Response: Dear reviewer, we sincerely appreciate your enthusiasm for our manuscript’s subject and hypotheses. Additionally, we value your detailed perspectives and insightful comments.

Methodology

1. The wording of second line in section data source, setting and sampling seems need correction.

Response: Dear reviewer, I sincerely appreciate your valuable insights. In response to your feedback, we have diligently made the necessary corrections. I kindly invite you to refer to the relevant section in our revised manuscript for further details.

2. Mentioning how researchers came with those potential predictors in section variables? Referencing again basically. I believe they are 7-11.

Response: Dear reviewer, I sincerely appreciate your valuable insights regarding variable selection in our study. In response to your suggestion, we have included how we selected potential predictors and provided references that informed our decision-making process regarding explanatory variables. Please find these details in the variable section of our revised manuscript.

3. The description provided for the expression of the generic model presented in section The Regression Model need amendments? Basically, correcting subscripts.

Response: Dear reviewer, Thank you for your insightful feedback regarding the model formulation. We have carefully considered your point and made the necessary amendments to the model. Please refer to our revised manuscript for the specific details.

4. In the regression model what assumptions were made? I see three random terms there. What distribution they have? Is r_jk independent of u_ik?

Response: Dear reviewer, Thank you for your valuable scientific inquiry. In our nested model with three levels, we operate under the following assumptions: Variations at different levels are intricately linked to the levels above them. At the foundation, Level 1, individual observations or measurements—variations in stunting among children—serve as the fundamental building blocks for our entire model. As we ascend to Level 2 (communities), we encounter community-level variations influenced by correlations between individuals within the same community. These community-level dynamics, in turn, reverberate upward to impact the highest level (Level 3), which represents countries. Importantly, our model explicitly accounts for the hierarchical structure inherent in the data: individuals nest within communities, and communities nest within countries.

5. Since sample size is large will considering a 0.01 level make sense? So, 99% CI needs to be calculated instead of 95% CI.

Response: Dear reviewer, Thank you for your thoughtful question. While the sample size in our study is indeed large, it’s essential to consider the practical context. However, when compared to the total population of low and lower-middle income African countries, our sample may not be adequate to confidently report a 99% confidence interval (CI). In the literature, similar studies have consistently reported findings using a 95% CI, which is widely accepted as the standard. Therefore, we recommend adhering to the 95% CI for consistency and comparability.

6. Did authors do data cleaning? If so, what was their procedure?

Response: Dear reviewer, Thank you for your insightful feedback on our manuscript. We sincerely appreciate your thorough review and valuable suggestions. We have incorporated a detailed explanation of our data cleaning procedures into the revised methodology section of the manuscript. This update ensures transparency and allows readers to understand the steps taken to prepare the dataset for analysis. We believe that these enhancements significantly strengthen the scientific rigor of our study. Kindly refer to the data cleaning and management section of our revised manuscript. 

Results

7. With table 2 researchers selected model IV. Is there a need for mentioning results from other models in table 3? 

Response: Dear reviewer, Thank you for your thoughtful question. Including results from other models in Table 3 is valuable for contextual relevance, transparency, and guidance for readers. By presenting a comprehensive view of different modeling choices, we enhance the robustness of our findings and facilitate meaningful discussions. This approach aligns with standard reporting practices in the scientific literature. 

(Refer: https://doi.org/10.1371/journal.pone.0295289, https://bmcmedresmethodol.biomedcentral.com/articles/10.1186/s12874-019-0876-8)

8. After considering 0.01 level, I believe there will more variables that will be insignificant. In that case will keeping only the significant variables in the model make sense?

Response: Dear reviewer, Thank you for your insightful question. When considering a stricter significance level (0.01), it’s essential to evaluate the practical context. While some variables may become insignificant, we recommend assessing their theoretical relevance. If a non-significant variable has practical importance or contributes to interpretability, it may still be valuable to retain it. The principle of parsimony suggests balancing statistical rigor with meaningful predictors. Additionally, retaining significant variables at the standard 0.05 level is reasonable.

9. If authors do not agree with 0.01 level, will keeping only significant variable in the model make sense?

Response: Thank you for your inquiry. While retaining significant variables is standard practice, we must balance statistical rigor with practical relevance. If a non-significant variable contributes to understanding or aligns with existing literature, it may be valuable to retain it. Authors’ agreement on the significance level is crucial, considering that some fields tolerate a 0.05 level due to practical constraints. Transparent reasoning remains essential.

10. “At the individual level, the odds of stunting were 1.76 times higher among children aged 24-59 months (AOR = 1.76 at 95% CI: 1.70, 1.83) compared to children aged less than 24 months.” In my opinion, this is not correct. The correct interpretation would be “At the individual level, the odds of stunting were 1.76 times as high among children aged 24-59 months (AOR = 1.76 at 95% CI: 1.70, 1.83) compared to children aged less than 24 months.” If authors agree, please change rest of the interpretations in this way.

Response: Dear reviewer, Thank you for your scientifically sound and valuable corrections regarding the interpretation of our results. We have diligently addressed the points you raised and made the necessary revisions in our manuscript. Kindly refer to the same section in our revised version. 

We sincerely thank the anonymous reviewers and the editor for their constructive comments and suggestions!

---

## [Decision Letter · Decision Letter 1]

1 Apr 2024

Prevalence of Childhood Stunting and Determinants in Low and Lower-Middle Income African Countries: Evidence from Standard Demographic and Health Survey

PONE-D-24-04373R1

Dear Dr. Tadesse Tarik Tamir,

We’re pleased to inform you that your manuscript has been judged scientifically suitable for publication and will be formally accepted for publication once it meets all outstanding technical requirements.

Kind regards,

Satyajit Kundu

Academic Editor

PLOS ONE

Reviewers' comments:

Reviewer #1: All comments have been addressed

2. Is the manuscript technically sound, and do the data support the conclusions?

Reviewer #1: Yes

3. Has the statistical analysis been performed appropriately and rigorously? 

Reviewer #1: Yes

4. Have the authors made all data underlying the findings in their manuscript fully available?

Reviewer #1: Yes

5. Is the manuscript presented in an intelligible fashion and written in standard English?

Reviewer #1: Yes

6. Review Comments to the Author

Reviewer #1: Thank you to the authors for addressing all my comments and the changes made to the manuscript. I have no additional comments.

7. PLOS authors have the option to publish the peer review history of their article (what does this mean?). If published, this will include your full peer review and any attached files.

Reviewer #1: No
